# A multiomic approach to defining the essential genome of the globally important pathogen *Corynebacterium diphtheriae*

Emily C. A. Goodall[1]☯*, Camila Azevedo Antunes[2,3]☯, Jens Möller[3]☯, Vartul Sangal[4], Von Vergel L. Torres[1], Jessica Gray[1], Adam F. Cunningham[5], Paul A. Hoskisson[6], Andreas Burkovski[3], Ian R. Henderson[1]*

1 Institute for Molecular Bioscience, University of Queensland, Brisbane, Australia, 2 Microbiome and Cancer Division, German Cancer Research Center (DKFZ), Heidelberg, Germany, 3 Microbiology Division, Friedrich-Alexander-Universität Erlangen-Nürnberg, Erlangen, Germany, 4 Faculty of Health and Life Sciences, Northumbria University, Newcastle upon Tyne, United Kingdom, 5 Institute of Immunology and Immunotherapy, University of Birmingham, Birmingham, United Kingdom, 6 Strathclyde Institute of Pharmacy and Biomedical Sciences, University of Strathclyde, Glasgow, United Kingdom

☯ These authors contributed equally to this work.

* e.goodall@uq.edu.au (ECAG); i.henderson@uq.edu.au (IRH)

**Data Availability Statement:** Transposon sequencing data are available for download at the European Nucleotide Archive (accession:

## Abstract

Diphtheria is a respiratory disease caused by *Corynebacterium diphtheriae*. While the toxin-based vaccine has helped control outbreaks of the disease since the mid-20th century there has been an increase in cases in recent years, including systemic infections caused by non-toxigenic *C. diphtheriae* strains. Here we describe the first study of gene essentiality in *C. diphtheriae*, providing the most-dense Transposon Directed Insertion Sequencing (TraDIS) library in the phylum Actinobacteriota. This high-density library has allowed the identification of conserved genes across the genus and phylum with essential function and enabled the elucidation of essential domains within the resulting proteins including those involved in cell envelope biogenesis. Validation of these data through protein mass spectrometry identified hypothetical and uncharacterized proteins in the proteome which are also represented in the vaccine. These data are an important benchmark and useful resource for the *Corynebacterium*, *Mycobacterium*, *Nocardia* and *Rhodococcus* research community. It enables the identification of novel antimicrobial and vaccine targets and provides a basis for future studies of Actinobacterial biology.

## Author summary

*Corynebacterium diphtheriae* causes both toxin-mediated diphtheria and non-toxigenic invasive infections. Despite a vaccine to protect against diphtheria, case numbers for both invasive and diphtherial disease have increased over the last decade. Furthermore, an increase in antibiotic resistant strains are being isolated from patients. It's clear that additional treatment strategies for this organism will be needed in the future. Using high-throughput mutagenesis, this work presents the densest library of mutants for any

PRJEB56349). The genome annotation used in this analysis is available for download here: 10.6084/m9.figshare.22332187. The processed insertion data can be viewed online at: https://tradis-vault.qfab.org/. Proteomics data are available via ProteomeXchange PXD036352.

**Funding:** PAH would also like to acknowledge funding from Medical Research Scotland (422 FRG) and the Royal Academy of Engineering Research Chair Scheme for long-term personal research support (RCSRF2021\11\15); PAH received a salary from RAEng for this work. The funders had no role in study design, data collection and analysis, decision to publish, or preparation of the manuscript.

**Competing interests:** The authors declare they have no competing interests.

*Corynebacterium* sp.. This work identifies the essential genome of *C. diphtheriae*; an important classification as these genes are often the target of therapeutic intervention. We identify highly conserved genes and species-specific genes unique to pathogens. This data presents an important benchmark and focus for the future development of therapeutic options. Of particular significance is the identification of uncharacterized, conserved proteins within the Diphtheria vaccine.

## Introduction

*Corynebacterium diphtheriae* is a globally important pathogen and the causative agent of diphtheria, an upper respiratory tract infection that is mediated by the corynephage-encoded diphtheria toxin [1]. Prior to the introduction of the vaccine, *C. diphtheriae* was responsible for significant global childhood mortality. Since the introduction of a vaccine in the mid-20th century, which neutralizes the diphtheria toxin, the number of fatal diphtheria cases fell dramatically [2,3]. However, it remains capable of rapid resurgence if vaccine regimens are disrupted by the breakdown of healthcare provision or large population displacements [4,5]. Approximately 5,000 cases of diphtheria were reported by the World Health Organization in 2008 increasing to almost 23,000 cases in 2019 [2,3]. Whilst the toxin is considered the major virulence factor in *C. diphtheriae*, in recent years several other virulence factors have been identified that act independently of the toxin [6–14]. Indeed, an increasing number of systemic infections caused by non-toxigenic *C. diphtheriae* strains, such as endocarditis, septic arthritis and osteomyelitis, have been reported across the globe [8,15–19]. Furthermore, a growing number of *Corynebacterium* species have been reported to cause opportunistic infection [20–24], or disease in animals [25], with at least 67 species being of clinical or veterinary importance [26].

The *Corynebacterium* genus belongs to the phylum Actinobacteriota; a phylum of diverse Gram-positive bacteria that includes the aerial mycelium- and spore-forming soil bacteria *Streptomyces*, the probiotic commensal *Bifidobacterium* and the human pathogen *Mycobacterium tuberculosis* [27]. In addition to the thick peptidoglycan layer, a characteristic of Gram-positive bacteria, the cell envelope within the order *Corynebacteriales* is characterized by an additional arabinogalactan polysaccharide layer covalently linked to the peptidoglycan and esterified to an external "mycomembrane" layer of mycolic acids [28,29]. The mycomembrane is unique to the order *Corynebacteriales* (reviewed extensively: [28–32]), which encompasses the CMNR genera (*Corynebacterium*, *Mycobacterium*, *Nocardia* and *Rhodococcus*) and includes notable human pathogens. The mycomembrane layer has drawn analogous comparisons to the outer membrane of Gram-negative bacteria, although chemically and genetically distinct [33]. This additional selective permeability barrier confers intrinsic resistance to antibiotics and detergents, and mediates interaction with host cells [28,34–36].

In recent years many whole genome sequencing (WGS)-based surveillance studies have been published in an effort to monitor localized outbreaks [37–44]. While *C. diphtheriae* has not historically been associated with antimicrobial resistance (AMR), recent WGS studies have detected an increasing prevalence of AMR genes [42–44]. Indeed, a novel mechanism of β-lactam resistance was recently identified in a fatal case of diphtheria [45].

Given the increase in both diphtheria and non-toxigenic infections caused by *C. diphtheriae*, coupled with the increasing detection of AMR genes and the disruption of vaccination programs caused by global instability attributed to the outbreak of CoVID-19 in addition to global food shortages and war, prompting population displacement, it is imperative to further

our understanding of the biology of this global pathogen. Genes essential for growth are often the target of antibiotics and host-defence systems, and therefore may represent novel intervention strategies to combat *C. diphtheriae*-mediated disease. However, essentiality studies of the *Corynebacterium* genus to date have either focused on non-pathogenic species, or predictive results from *in silico* data [46,47]. To identify putative antibiotic and vaccine targets, we constructed the first high-density transposon library in the human pathogen and non-toxigenic clinical strain *C. diphtheriae* ISS 3319 using Transposon-Directed Insertion-site Sequencing (TraDIS) [48]. This Transposon Insertion Sequencing (TIS) technique has been used to identify genes essential for viability in diverse pathogenic and multidrug resistance bacteria [48–56]. Our high-resolution library has enabled the identification of essential gene and protein-coding domains in *C. diphtheriae*. Coupling these analyses with mass spectrometry-based proteomics enabled verification of expression of highly conserved, essential hypothetical proteins of unknown function. This work provides a benchmark for the comparison of gene essentiality among pathogenic and non-pathogenic *Corynebacterium* species and provides a starting point for understanding the biology of an important human pathogen, and expands our understanding of mycolate-containing species and the wider Actinobacteriota.

## Results and discussion

### Construction of a transposon insertion library in *C. diphtheriae* ISS 3319

While diphtheria is caused by toxigenic strains, the non-toxigenic *C. diphtheriae* can cause severe invasive infection due to a range of virulence factors [7,11,36,57–61]. We therefore chose to investigate the physiology of the non-toxigenic *C. diphtheriae* strain ISS 3319 due to its clinical importance in addition to the availability of a closed reference genome and published phenotypic studies. Moreover, the genetic tractability and easy culture conditions of this organism give it credence as a model for the study of mycolated pathogens.

Using previously described methods [56,62], we constructed a transposon mutant library in *C. diphtheriae* strain ISS 3319 using a mini-Tn5 transposon as they are considered to have negligible insertion bias for the analysis of essential genes [63]. Approximately $6\times10^5$ independent mutants were pooled to form the transposon library and two independent aliquots were sequenced (S1 Fig). Data were demultiplexed by barcode, then both the barcode and transposon sequences were matched and trimmed. The remaining reads were mapped to the reference strain (accession: CP025209.1). The insertion index scores (IIS), which represents the density of transposon insertions per coding sequence normalized by gene length, were calculated for each gene and these scores were compared between technical replicates (S2A Fig). The correlation coefficient between the IISs of the technical replicates was 0.95 (S2A Fig), therefore sequence data was combined resulting in a total of >4 million mapped reads and 206,873 unique insertions sites identified (Table 1). For a genome size of 2,404,936 bp (JGI ID: Go0113969), encoding 2,257 proteins, this equates to an average of one insertion every 12 bp (Table 2). A handful of transposon libraries have been constructed in *Corynebacterium* species [64–70], however, only one dense library has been constructed: in the industrial strain *C. glutamicum* MB001 [47]. Our library represents the first dense transposon library in pathogenic *C. diphtheriae*, and the densest library within the *Corynebacterium* genus (Fig 1A and Table 2). The insertion data are available to view at our online browser: https://tradis-vault.qfab.org/.

### Identification of the essential genome of *C. diphtheriae*

To identify essential genes, we calculated the IISs from the combined technical replicate data, as described previously [48,50,56,71,72]. The IISs displayed a bimodal distribution, with essential (low number of transposon insertions) and non-essential (considerably high transposon

**Table 1. Parameters for TraDIS data set derived from the non-toxigenic *C. diphtheriae* strain ISS 3319 transposon library.**

|  | Inline barcode match[a] | Transposon check 1[a] | Transposon check 2[a] | Mapped reads | Number of unique insertions |
|---|---|---|---|---|---|
| *C. diphtheriae* ISS 3319 genome | 5,946,186 | 5,633,492 | 5,215,460 | 4,256,044 | 206,873 |
| Rep1 | 3,319,966 | 3,066,609 | 2,826,453 | 2,233,957 | 159,937 |
| Rep2 | 2,626,220 | 2,566,883 | 2,389,007 | 2,022,087 | 171,915 |

[a] These barcodes/tags correspond with the sequential pattern matching and trimming positions of the raw sequencing data during data processing. Please refer to S15B Fig for further detail.

insertions) genes associated with the left and right mode, respectively (S2B Fig). The probability of belonging to each mode was calculated, and the ratio of these values was termed the log-likelihood score with a threshold of $\log_2$ (12) (12 times more likely to belong to one mode than the other mode) [48,50,56,71,72]. We used a conservative, and more stringent, threshold of 12× to preserve confidence in the list of predicted essential genes. Genes with a log likelihood score ≤ -3.6 were classified as essential, genes with a log likelihood score ≥3.6 were classified as non-essential, and those with a log likelihood score between these values were deemed 'unclear' [48]. This analysis indicated that 1,801 genes in the library possessed sufficient insertions to suggest they were non-essential for growth on HI agar plates, with 341 genes classified as essential and 115 genes categorized as unclear (S1 Table and S2B Fig).

For certain essential genes transposon insertions can be tolerated within specific locations corresponding to nonessential domains of the encoded protein. Such genes can have sufficiently high insertion index scores that they do not meet the computational threshold of insertion density to be classified as 'essential'. To overcome this limitation, we computed the probability of observing a stretch of uninterrupted genome, and therefore a likely essential region. Using a previously described method [56], which assumes random insertion of the transposon, we calculated that for a 2,404,936 bp genome with a library density of 206,873 unique insertions, the probability of observing, an insertion-free region (IFR) of 178 bp is significant (Threshold $p = 0.05$) within a genome, and an IFR of 74 bp is significant (Threshold $p = 0.05$) within a gene (S3 Fig). Note the smallest annotated gene (*diphtheriae_01371*) is 90 bp and larger than the size threshold of 74 bp, therefore no annotated genes were excluded from our downstream analysis. The 1,916 genes that did not meet our initial criteria of essentiality were re-analyzed: calculating the proportion of insertion-free regions relative to the CDS

**Table 2. Transposon mutant libraries of the *Corynebacterium* genus.**

| Strain | Year | No. of mutants[a] | Genome size (bp) | Insertion density[b] | Reference |
|---|---|---|---|---|---|
| *C. diphtheriae* C7(−) | 2002 | ~3,500 | 2,499,189 | 714.05 | [64] |
| *C. matruchotii* ATCC 14266 | 2003 | ~33,000 | 2,866,540 | 86.86 | [66] |
| *C. pseudotuberculosis* T1 | 2006 | 1,500 | 2,337,578 | 1558.38 | [67] |
| *C. glutamicum* ATCC 13032 | 2006 | 10,080 | 3,282,708 | 325.67 | [68] |
| *C. glutamicum* R | 2006 | 11,241 | 3,363,299 | 299.20 | [65] |
| *C. glutamicum* MB001 | 2019 | 200,940 | 3,079,253 | 15.32 | [47] |
| *C. glutamicum* 2262 | 2020 | 10,073 |  |  | [69] |
| *C. glutamicum* ML103 | 2021 | 3,207 |  |  | [70] |
| *C. diphtheriae* ISS 3319 | 2022 | 206,873 | 2,404,936 | 11.63 | This study |

[a] Where both the number of colonies and number of detected insertion sites were available, the total number of insertion sites was used.

[b] Insertion density is the genome size divided by the number of insertions or mutants reported for that genome to give an approximation of 1 insertion every X bp as a measure of overall library density.

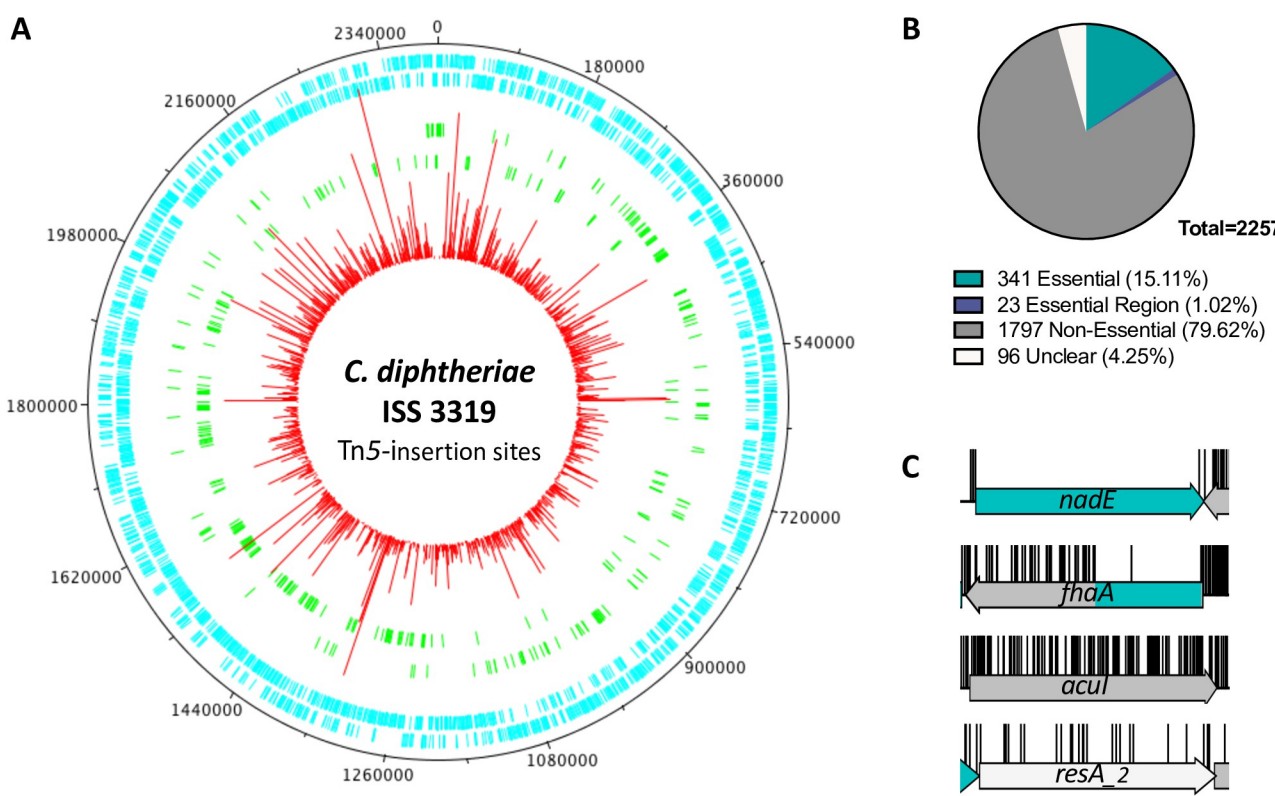

**Fig 1. Comprehensive genome transposon insertion sites mapped to *C. diphtheriae* strain ISS 3319.** (A) Frequency and location of the transposon junction sequences from the Tn5 transposon library in the *C. diphtheriae* strain ISS 3319, mapped to the ISS 3319 genome (CP025209.1). The external track marks the ISS 3319 genome in bp starting at the annotation origin. The next two inner tracks correspond to sense and anti-sense CDS respectively (light blue), followed by two inner tracks depicting the essential genes identified by TraDIS on the sense and anti-sense strands, respectively (green). The innermost circle (red) corresponds to the frequency and location of transposon insertion sequences mapped successfully to the ISS 3319 genome after identification of a transposon sequence. Figure created using DNAPlotter. (B) Proportion of essential genes for the genome. (C) Representative insertion profiles of an essential gene (green; *nadE*), a domain-essential gene (*fhaA*); a non-essential gene (grey, *acuI*) and a gene classified as unclear (white; *resA_2*).

length as a percentage, revealed an additional 23 domain-essential genes with insertion-free regions ≥ 40% of the CDS (Figs 1B, S4 and S2 Table). This enabled revision of the total number of essential genes to 364 and unclear genes to 96 (Fig 1C). Besides essential domains within a CDS, there are a number of additional explanations for lowering the IIS of a gene (discussed extensively elsewhere [56]), such as (1) if a gene, when disrupted, has a severe fitness cost, such that the mutant is slow-growing or forms a small colony on a plate, resulting in the mutant being in low abundance relative to other non-essential mutants within the total mutant pool. (2) If a gene can only tolerate transposon insertions for one orientation of the transposon, for example as a consequence of polar effects, resulting in half as many possible insertions and lowering the IIS. (3) If physiological conditions limit transposon insertion, such as DNA-binding proteins (discussed later). However, in all these scenarios these genes are non-essential and we did not investigate the "unclear" genes further.

## Sub-CDS resolution reveals an extended essential domain in the lipid II flippase, MviN

Domain essentiality analysis revealed an unusual insertion profile (of alternating essentiality predictions) within the gene *mviN* (*DIP2371*). The gene *mviN* encodes the lipid II flippase,

which is encoded by the MurJ domain of the protein, required for peptidoglycan synthesis and essential in all bacteria. In our data we observed Cdip-MviN possessed an extended C-terminal domain with sequence homology to the *M. tuberculosis* MviN homolog Rv3910 (Fig 2A; [73]). Our data also indicate that besides the essential MurJ domain there are two large IFRs (343 bp and 119 bp) that coincide with the putative cytoplasmic domain of the protein, suggesting this may also be essential in *C. diphtheriae* (Fig 2B). These data were unexpected for two reasons: (1) the extended C-terminal domain in Mtb-MviN encodes a pseudokinase domain and an extracytoplasmic domain that regulate peptidoglycan synthesis, and to date this mechanism of regulation has only been reported in *Mycobacterium* spp. [74], and (2) in *Mycobacterium* spp. these domains have been shown to be dispensable by both transposon mutagenesis data and domain-deletion studies [74]. In *Mycobacterium*, an essential serine/threonine protein kinase (STPK), PknB, regulates cell wall synthesis via the phosphorylation of Thr$^{947}$ in the pseudokinase domain of MviN [74–76]. This in turn recruits the fork-head associated (FHA) domain of FhaA (S5A Fig). FhaA regulates peptidoglycan synthesis via interaction with PbpA [74].

Sequence comparisons showed conservation of the phosphorylated threonine residue in MviN of *C. diphtheriae* (Thr$^{947}$ in Mtb-MviN, Thr$^{918}$ in Cdip-MviN: Fig 2C). Modelling the Cdip-MviN cytoplasmic domain structure also shows similarity to the structure of the pseudokinase domain of Mtb-MviN (PDB 3OUN; Fig 2C and 2D). Moreover the Ser$^{473}$ and Arg$^{474}$ residues of Mtb-FhaA that interact with P-Thr$^{947}$ of Mtb-MviN are conserved in *C. diphtheriae* (S5D Fig). Together these data suggest that the interaction between MviN and FhaA is conserved in *C. diphtheriae* and suggest a potential mechanism of regulation of peptidoglycan synthesis in *C. diphtheriae*.

However, the species have opposing essentiality requirements within the PknB-MviN-FhaA pathway (S5 Fig). While PknB is essential in *Mycobacterium*, our data suggest that *pknB* (*DIP0053*) is not essential, consistent with earlier data from *C. glutamicum* (S5B Fig) [77]. In support of the TIS data we constructed an isogenic *pknB* mutant and confirmed its growth and genotype (S5C and S5D Fig). Our data also suggest that the pseudokinase domain of Cdip-MviN is essential, in contrast to *Mycobacterium*. Furthermore, FhaA is non-essential in *M. tuberculosis* and *M. smegmatis* (although deletion results in morphology defects [74,78]). In our data, the putative phosphopeptide-binding domain (DUF3662) of *fhaA* (*DIP0059*) is essential while the FhaA FHA domain appears to be dispensable (S5E and S5F Fig). These data highlight genus-specific differences in the regulation of peptidoglycan synthesis, which is further emphasized by the observation that the extended pseudokinase domain of MviN only appears to be present in organisms that possess an arabinogalactan layer within their envelope (S6 Fig) suggesting that coordination of cell envelope biogenesis in the mycolata is highly organized and appears to have genus specific requirements.

## Conservation of essential genes in corynebacteria

To determine the conservation of the 364 essential genes within the *Corynebacterium* genus, using BlastP we searched for homologs of the essential genes within a dataset of 140 curated *Corynebacterium* genomes representing 126 species [26](S3 Table). Where more than one match was identified per genome, the match with the best E-value was used. Of these 364 genes, 358 returned a match in at least one genome (Figs 3A and S7 and S4 Table). The six annotated genes that had no homologs in any other genome were found to be annotation artefacts and we revised our essential gene list to 358 essential genes (S2 Table). Of these, 345 were detected in >75% of the genomes (Fig 3B), eight were moderately conserved (detected in 25–75% genomes) and five were poorly conserved (<25% genomes). Of the five narrowly distributed genes (Fig 3B): *diphtheriae_01987* (*DIP2072*); *diphtheriae_00762* (*DIP0858*);

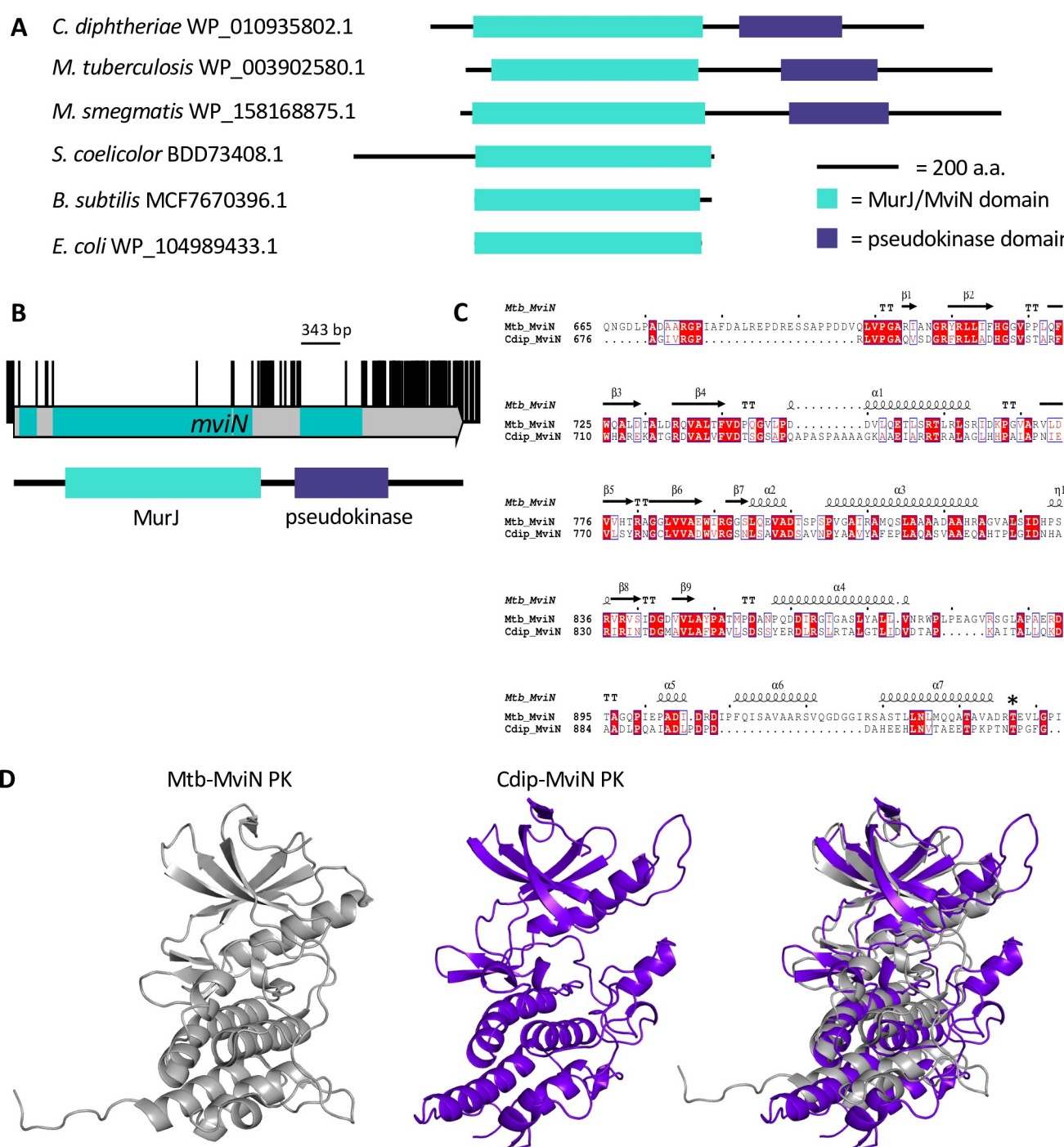

**Fig 2. Essential domains of MviN.** (A) Schematic alignment of MurJ orthologs from *Corynebacterium diphtheriae*, *Mycobacterium tuberculosis*, *Mycobacterium smegmatis*, *Streptomyces coelicolor*, *Bacillus subtilis* and *Escherichia coli*. (B) Transposon insertion data of *mviN*. Transposon insertion sites represented by black vertical bars, capped at a frequency of 1. Essential regions of the coding sequence that coincide with insertion-free regions are highlighted in green within the gene arrow. Protein domain boundaries are plotted below: MurJ lipid-II flippase domain (green), pseudokinase domain (purple). (C) Pairwise sequence comparison of the cytoplasmic pseudokinase domains of Mtb-MviN with Cdip-MviN. Conserved residues are highlighted in red, P-Thr$^{947}$ indicated by an asterisk (*), with the secondary structure of Mtb-MviN displayed above. (D) Comparison of the crystal structure of the Mtb-MviN pseudokinase domain (grey) and the predicted model of Cdip-MviN pseudokinase domain (purple) from Phyre2.

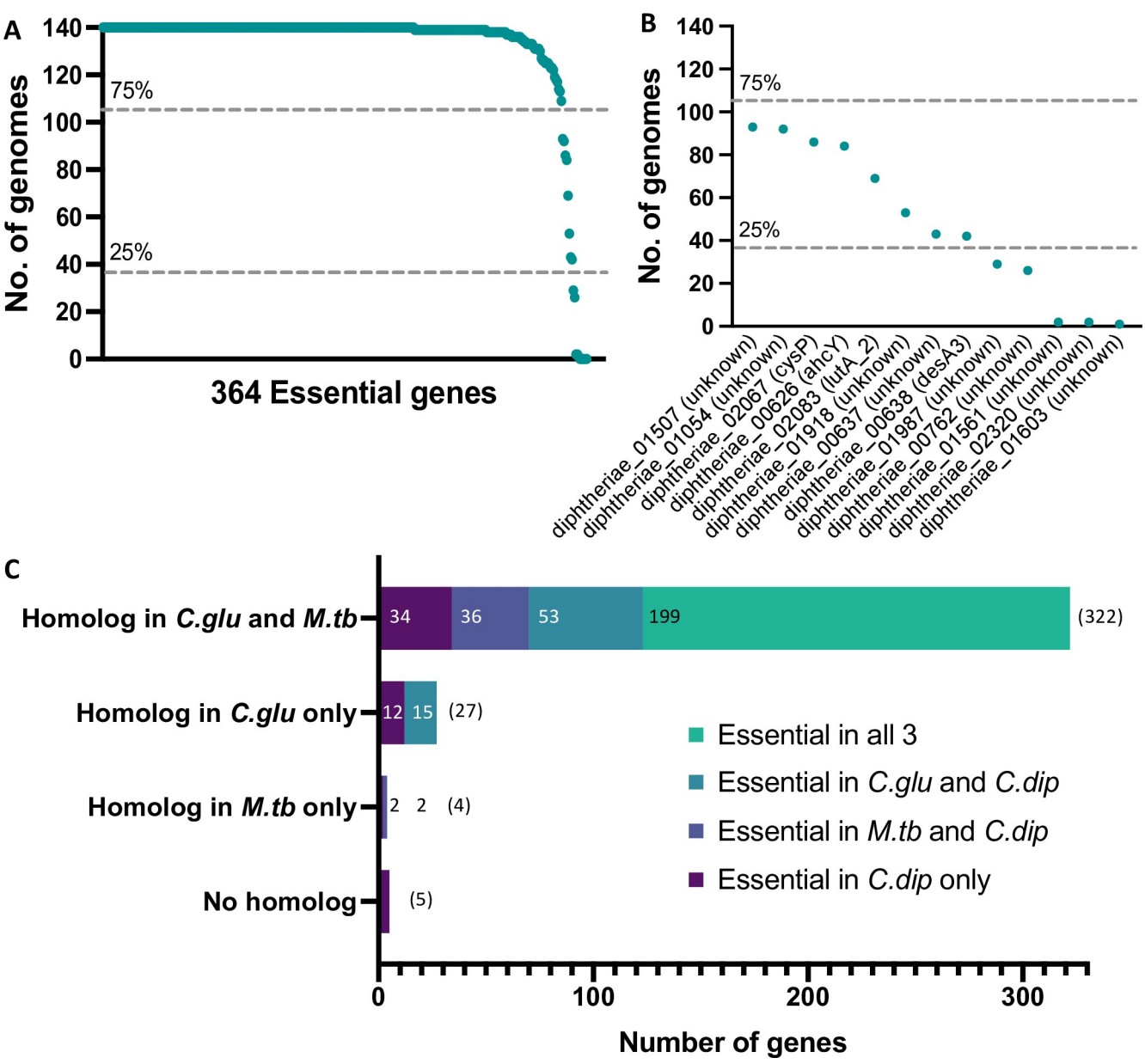

**Fig 3. Orthologs of the essential genes of *C. diphtheriae*.** (A) Frequency plot of essential gene homologs within 140 genomes of representative *Corynebacterium* species. (B) Panel derived from [A] depicting the subset of genes conserved in fewer than 75% of the representative *Corynebacterium* genomes. (C) Conservation of the candidate 358 essential genes of *C. diphtheriae* in representative strains of *M. tuberculosis* H37Rv and *C. glutamicum* MB001 with equivalent transposon insertion sequencing datasets. Genes are grouped into categories of shared homology (E-value ≤1e-5, % identity ≥30 and % coverage ≥30) and coloured according to gene essentiality information.

*diphtheriae_02320*; *diphtheriae_01561*; *diphtheriae_01603*), three genes (*diphtheriae_02320*; *diphtheriae_01561*; *diphtheriae_01603*) were highly restricted to *C. diphtheriae* and the closely related species *C. belfantii*. These three genes are also conserved in *C. diphtheriae* NCTC 13129, however, the genes are not annotated in this strain explaining why they were not identified in the initial BlastP screen. Despite the lack of functional information for these genes, the transposon insertion data suggests these genes are important. The CDS diphtheriae_01987 C-terminus is predicted to contain an AbiEi antitoxin domain and appears sporadically

distributed throughout the genus. While the CDS diphtheriae_00762 is a hypothetical protein that appears restricted to phylogroups G, H and Q, which contain the majority of the *Corynebacterium* sp. clinical isolates [26].

Finally, we applied hierarchical clustering to the matrix of homologs to view whether there were any shared conservation patterns between different genes across the genus. Unsurprisingly, given the variation between cell wall composition in *Corynebacterium*, genes associated with mycolic acid and cell wall biosynthesis are not completely conserved across the genus. Examples of this differential conservation include the methylerythritol phosphate biosynthetic pathway (required for isoprenoid biosynthesis), the diaminopimelate (*m*DAP) biosynthetic pathway, where variation in *Corynebacterium* mDAP is known [79], and the NADPH oxidoreductase (*diphtheriae_00637; DIP0703*) and NADPH-dependent stearoyl-CoA 9-desaturase (*desA*3; *DIP0704*) operon that is likely involved in double-bond formation in corynemycolates (S8 and S9 Figs).

## Comparison of essential genes of *C. diphtheriae* to *C. glutamicum* and *M. tuberculosis*

We extended our analyses to compare all the essential genes of *C. diphtheriae* to equivalent essential gene datasets reported for two other mycolated organisms: *C. glutamicum* and *M. tuberculosis* [47,55]. The 358 genes predicted to be essential in *C. diphtheriae* were compared against the total protein CDSs of both *C. glutamicum* MB001 and *M. tuberculosis* H37Rv using blastp [80]. Of the 358 essential genes of *C. diphtheriae*, 322 orthologs were identified in both *C. glutamicum* and *M. tuberculosis*, with the majority (199) essential in all three species. In contrast, 34 genes were conserved but essential only in *C. diphtheriae* (Fig 3C and S5 Table). Some differences between essentiality data can be attributed to subtle differences in the experimental conditions such as library construction method, choice of transposon, or analytical methods. For example, of the 322 shared orthologs, 10 were essential in *C. diphtheriae* and *C. glutamicum*, but due to their small size and lower density library were classified as 'uncertain' in the *M. tuberculosis* (S5 Table). These genes are predicted to encode small ribosomal proteins, which are likely essential in all species demonstrating the advantage of screening saturated transposon mutagenesis libraries with minimal insertion bias.

*C. diphtheriae* shared 27 essential orthologs exclusively with *C. glutamicum*, while only four genes have an ortholog in *M. tuberculosis* but are absent in *C. glutamicum*. Included in the *Corynebacterium*-specific essential genes was the *mrp* operon (S5 Table). The *mrp* operon encodes a hetero-oligomeric Na$^+$/H$^+$ antiporter and is found in a diverse selection of Gram-negative and Gram-positive bacteria. However, phylogenetic distribution varies widely, especially where alternate transporters are utilized within a group [81,82]. The physiological role of the Mrp complex varies between species: it is essential in *Bacillus subtilis* for Na$^+$ tolerance [83,84], it is required for arsenic resistance in *Agrobacterium tumefaciens* [85], for pH tolerance in *Sinorhizobium meliloti* and pathogenesis in *Pseudomonas aeruginosa* [86,87]. Our data suggests it is essential in *Corynebacterium* spp. under laboratory growth conditions and presents a unique transport pathway that is distinct from *M. tuberculosis* (S10A Fig).

Conversely, of the four genes shared with *M. tuberculosis*, two of these unique genes form an operon (*diphtheriae_00637* and *desA3*) encoding a NADPH oxidoreductase and NADPH-dependent stearoyl-CoA 9-desaturase respectively (see above) that was absent from *C. glutamicum*, and essential in *C. diphtheriae* only. Given that these enzymes insert double bonds into fatty acid chains, they may reflect key steps in the formation of *C. diphtheriae* specific elongated mycolic acids that are absent in *C. glutamicum* [88].

Of the remaining *Corynebacterium* specific genes, 10 of these are annotated as "unknown" genes which enables the prioritization of targets for further study. For those with annotated names, on first inspection *ftsL_2* (*diphtheriae_01542; DIP1605*) appeared to lack a homolog in *M. tuberculosis*. After closer examination *ftsL_2* shares gene neighbourhoods to FtsL orthologs in *C. glutamicum* and *M. tuberculosis*, and was initially missed in our analysis due to low sequence similarity consistent with previous reports in *Mycobacterium* [89]. Further investigation found *ftsL_1* (*diphtheriae_00817; DIP0918*) shares comparable gene neighbourhoods with the FtsB orthologs cgp_1112 and Rv1024 and has likely been misannotated in both our strain and *C. diphtheriae* strain NCTC 13129 (S10B Fig). As such, a limitation of sequence homology comparison is that proteins that share functionality but not sequence identity may be falsely classified, as is most likely the case for the three *atp* genes that apparently had no homolog. The nine remaining genes unique to *Corynebacterium* sp. did not appear to have overlapping function.

### Functional classification of essential genes

A limiting factor for the analysis of less well characterized genomes is poor annotation. Using previous functional analysis of the *C. diphtheriae* ISS 3319 strain [90], we reviewed the functional classification of its genes. We found that 25% (in number) of all genes were classed as "uncharacterized", while "uncharacterized" genes comprise ~6% of the essential genome. The largest share of essential genes were involved in "Information storage and processing" (which includes processes such as DNA replication, transcription and translation) and "Metabolism", consistent with equivalent datasets (Fig 4A and S6 Table). There was one essential gene (*diphtheriae_01478/DIP1546*) classified in the "Involved in pathogenesis" category, which is required for gut colonization in a *C. elegans* model [62]. Our transposon insertion sequencing data suggests that this gene is also important for viability under laboratory growth conditions and the resolution of this dataset suggest that the 5′ end of the CDS is essential, while the 3′ end can be disrupted.

We next reviewed the localization of the essential gene products, or essential proteins [58]. 74.86% of the essential proteins were predicted to be cytoplasmic, 14.25% were predicted to be transmembrane proteins and 6.98% were predicted to be secreted (Fig 4B); the specific predicted secretion pathways are shown in S11 Fig. The localization of the remaining proteins was ambiguous (3.35%) or unknown (0.56%). Overall, approximately one fifth of essential proteins are either envelope associated or secreted, however, the secreted and membrane-associated essential proteins are the least well characterised (Fig 4C and S7 Table).

Finally, we checked for conserved domains (CD) within the essential proteins using NCBI's CD-search tool [91]. In total, domains were identified for 347/358 proteins, with 17 proteins predicted to have >1 functional domain while no hits were identified for 11 proteins. Of note, nine domains of unknown function were identified within the essential proteome; two of which are predicted to be secreted and five are predicted to be TM proteins, further highlighting gaps in our understanding of essential cell-envelope or secreted protein biology (S8 Table). These present unexplored avenues for antimicrobial intervention.

### Validation of essential gene expression

Genes can be falsely classified as essential if they are heavily silenced by DNA-binding proteins, preventing transposon insertion and resulting in a low insertion index score. Such genes would therefore not be expected to be detected by proteomics analysis. To validate the expression of the essential genes in *C. diphtheriae* ISS 3319, we evaluated the total and secreted proteome by mass spectrometry. Briefly, cells were grown in BHI medium before harvesting both

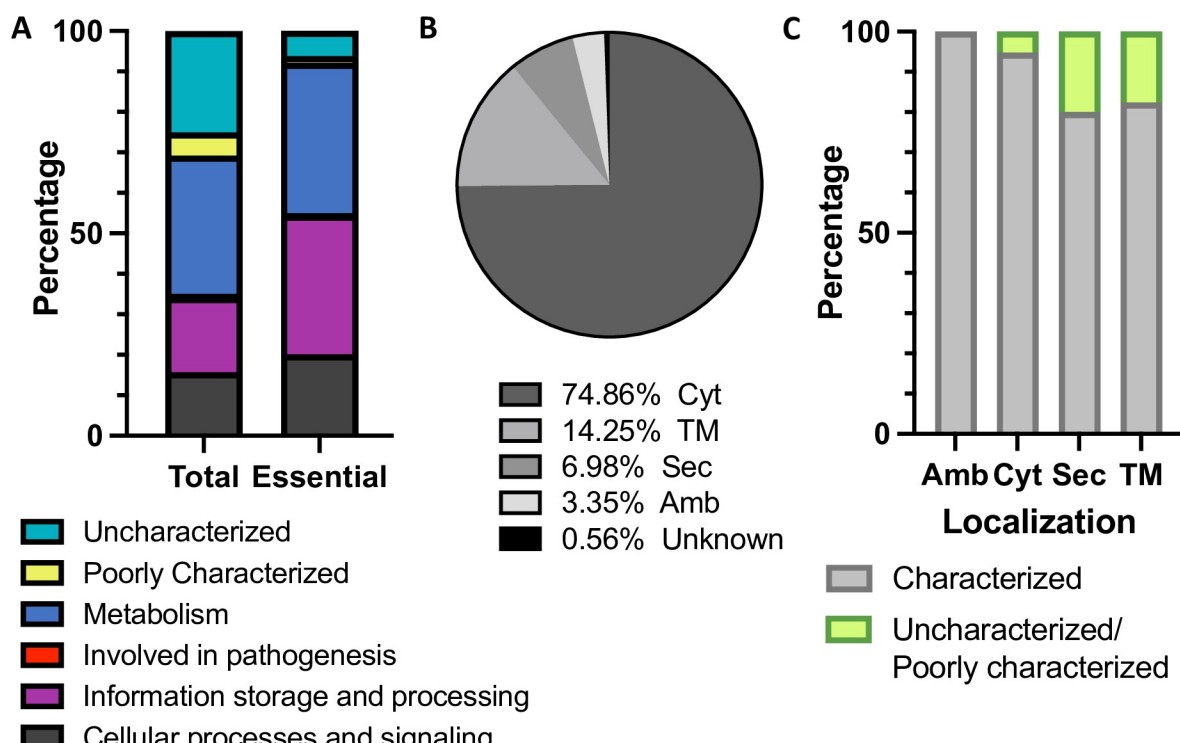

**Fig 4. Protein function and predicted localization.** (A) The number of proteins within each functional category for both the total genome and the list of essential genes. (B) The predicted localization of essential proteins. (C) Functional characterization of essential proteins by localization. Abbreviations: Cyt, cytoplasmic; TM, transmembrane; Sec, secreted; Amb, ambiguous.

the total cell fraction and a supernatant fraction. Samples were prepared as described in the materials and methods and subjected to mass spectrometry. A UniProt database was not available for *C. diphtheriae* strain ISS 3319 but does exist for the *C. diphtheriae* reference strain NCTC 13129 (accession BX248353), therefore, we compared the annotated protein CDSs of ISS 3319 with the annotations of NCTC 13129 using blastP (S9 Table). 2076/2257 (92%) annotated *C. diphtheriae* ISS 3319 genes had a homolog in *C. diphtheriae* NCTC 13129. The vast majority possessed ≥90% identity (1948/2076) and/or ≥90% coverage (1944/2076), therefore we deemed the NCTC 13129 proteome database suitable for analysis, with caveats highlighted below. Following mass spectrometry analysis, a total of 1,203 proteins were detected overall and 62 proteins were detected in the secreted fraction (S10 Table). We detected 286 of the 358 essential proteins, including diphtheriae_00762 and diphtheriae_01987 which were narrowly distributed within the corynebacteria (Fig 3B). Only two essential genes (*diphtheriae_01603* and *diphtheriae_02320*) did not have an annotation in *C. diphtheriae* NCTC 13129 (discussed previously) and were excluded from our proteomics analysis. However, of the 72 essential proteins that we did not detect, many of these, are associated with the cell envelope (for example MviN) and may have been removed with cell wall debris and membranes during sample preparation. Similarly, proteins of low abundance, or with post-translational modifications that impede trypsin digestion, may also be missed by our analysis. Of the 286 essential proteins we were able to validate by mass spectrometry, 11 had been previously predicted as hypothetical. Our analysis confirms they are expressed under standard laboratory conditions and predicted to be essential.

The proteomics analysis also enabled us to quantify protein abundance. By this measure, the largest share of the essential proteins from the cellular proteome was involved in

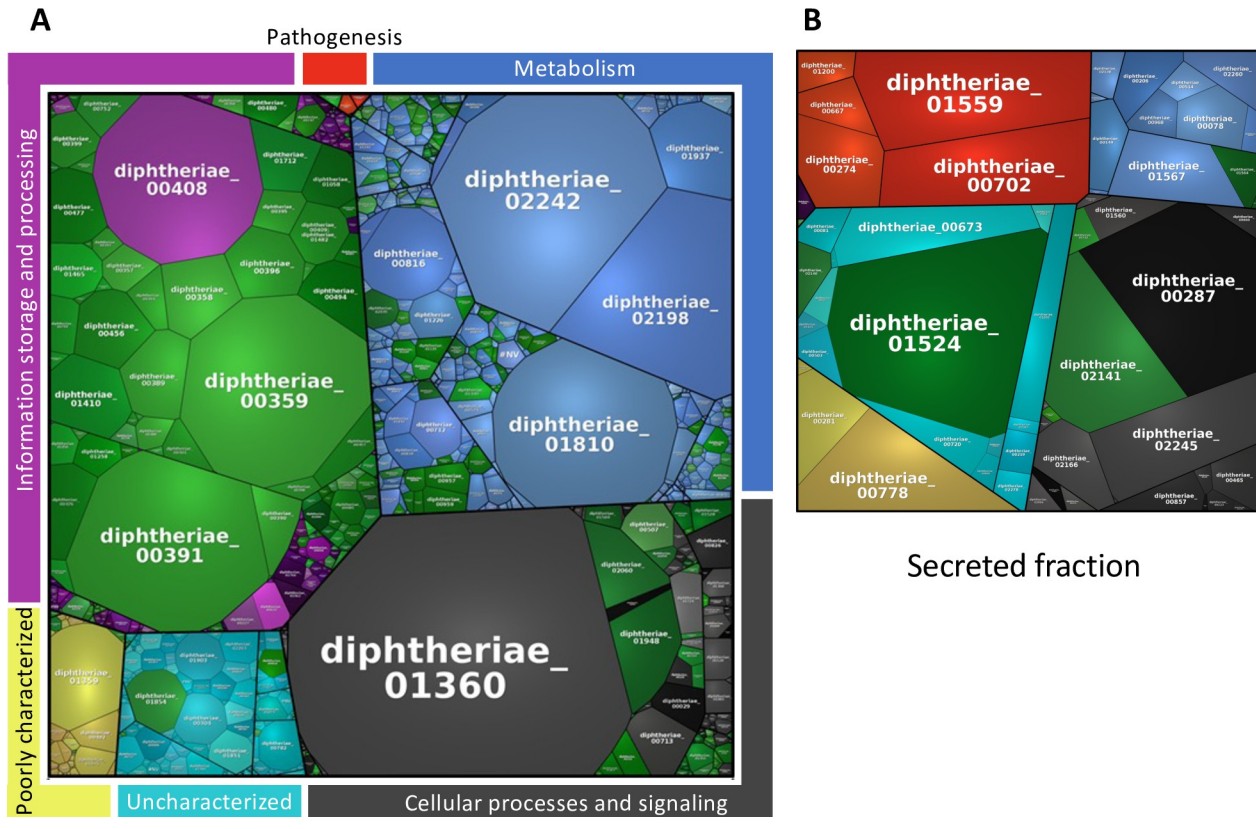

**Fig 5. Protein function and abundance plot of the whole and secreted proteome detected by mass spectrometry analysis.** The abundance of the total proteome (A) or secreted proteome (B) detected by mass spectrometry analysis grouped and coloured according to function, with block size corresponding with protein abundance. Essential proteins are highlighted in green.

information storage and processing (28.58%) followed by proteins involved cellular processes and signalling (3.94%), metabolism (3.47%), uncharacterized proteins (0.72%), poorly characterized proteins (0.06%) and proteins involved in pathogenesis (0.05%) (Fig 5A and Table 3). The two most abundant essential proteins were the translation-machinery associated proteins elongation factor Tu (diphtheriae_00391; DIP0470) and the 50S ribosomal protein L7/L12 (diphtheriae_00359; DIP0437), while the most abundant non-essential protein was an alkyl hydroperoxide reductase subunit C (diphtheriae_01360; DIP1420).

**Table 3. Functional abundance of proteins detected by mass spectrometry.**

| Biological process | Whole proteome | | Secreted proteins | |
|---|---|---|---|---|
| | Total [%][a] | Essential [%] | Total [%] | Essential [%] |
| Cellular processes and signalling | 24.29 | 3.94 | 28.98 | 6.69 |
| Information storage and processing | 35.33 | 28.58 | 4.19 | 3.96 |
| Involved in pathogenesis | 0.16 | 0.05 | 20.09 | 0.00 |
| Metabolism | 33.03 | 3.47 | 11.26 | 0.78 |
| Poorly characterized | 2.17 | 0.06 | 7.90 | 0.00 |
| Uncharacterized | 5.01 | 0.72 | 27.58 | 15.97 |
| Total | 100 | 36.84 | 100 | 27.40 |

[a] Percentages are calculated from measures of abundance normalized for protein size

There were nine essential proteins classed as "Uncharacterized" that were detected by mass spectrometry. The two most abundant of these uncharacterized proteins (diphtheriae_01854, DIP1918; diphtheriae_00818, DIP0919; Fig 5A) contain domains of unknown function (DUF), DUF501 and DUF3073 corresponding with pfam domains PF04417 and PF11273 (S12A and S12B Fig). Of note, the respective genes were also conserved and reported to be essential in both *C. glutamicum* and *M. tuberculosis* (*cgp_1113* and *Rv1025*; *cgp_2853* and *Rv0810c*, respectively; S13 Fig). Further phylogenetic analyses of these genes revealed they are highly conserved within the Class Actinomycetia (S12C Fig). The gene *diphtheriae_00818* is present in a putative conserved operon downstream of *ftsL_1* (FtsB; S13 Fig), which given the syntenic relationship with FtsB, it is possible that diphtheriae_00818 may have a role in cell division.

Finally, among the 62 proteins detected in the secreted fraction, 52 have previously been predicted to be secreted proteins with the remaining 10 classed as "ambiguous" [58], 20 are predicted to be secreted lipoproteins and 20 are predicted to be secreted with a SpI signal peptide (S10 Table). Eight essential proteins were detected in the secreted fraction (Fig 5B). The most abundant essential proteins were identified as uncharacterized proteins (15.97%), proteins involved in cellular processes and signalling (6.69%), proteins involved in information storage and processing (3.96%) and metabolism (0.78%). The most abundant non-essential secreted protein was diphtheriae_01559 (DIP1621), a homolog of the endopeptidase CgR_2070 (Uniprot accession: A4QFQ3) in *C. glutamicum* R [92]; while the most abundant essential secreted protein was the uncharacterized protein diphtheriae_01524 (DIP1586; Fig 5B). Overall, essential proteins made up 36.84% of the detected proteome, and 27.40% of the secreted fraction under laboratory conditions.

## Comparison of essential proteins with the Diphtheria vaccine

The diphtheria vaccine was recently found to contain other proteins besides the toxin [93]. The vaccine contained a total of 130–435 proteins depending upon the country of origin, with 65 proteins shared by all vaccines. We cross-referenced this data against ours and found 14 of the essential proteins have also been detected in six different global diphtheria vaccines and 12 of these are also reportedly immunogenic. Within the secreted fraction, three of the essential proteins have also been detected in the diphtheria vaccine (Möller et al. 2019) (diphtheriae_00615, DIP0680; diphtheriae_01524, DIP1586; diphtheriae_02141, DIP2193). These correspond with a putative anionic cell wall polymer biosynthesis enzyme, an uncharacterized protein and a putative mycolyl transferase, homologous to the antigen 85 proteins of *M. tuberculosis* and PS1 protein of *C. glutamicum* (S14 Fig) [29,58,94–96]. *M. tuberculosis* encodes three functional mycolyl transferases (*fbpA*, *fbpB* and *fbpC2*, named for their fibronectin-binding properties, which together constitute the antigen 85 complex [97]), while *C. glutamicum* encodes up to six mycolyl transferase homologs [31,98–100]. A trehalose analogue that inhibited mycolyltransferase activity *in vitro* has a bacteriostatic effect on the growth of *M. aurum*, suggesting the function of the mycolyltransferases is essential [101]. However, these proteins are functionally redundant, enabling disruption of individual loci [98,102]. Therefore, it was unexpected that *diphtheriae_02141* had both an essential domain, corresponding with the carboxylesterase domain, and a non-essential C-terminal domain in our TIS data (S14 Fig). One explanation is that the Fbp proteins of *C. diphtheriae*, with fewer *fbp* homologs than *C. glutamicum*, may not share the same functional redundancy [31], but this needs further investigation. Nevertheless, the presence of essential proteins within the diphtheriae vaccine that are immunogenic raises the possibility of the diphtheriae vaccine offering some cross protection to non-toxigenic *C. diphtheriae* strains.

## Conclusion

The rising case numbers of diphtheria and systemic infections caused by non-toxigenic *C. diphtheriae* worldwide mean it is important to expand our understanding of its biology to develop improved treatment strategies for the future. This is the first study to identify the total essential genome of the pathogenic species *C. diphtheriae*, a pathogen of historical significance. The sub-CDS level of resolution offered by our data enables us to define the essential genome and specific domains of proteins required for essential function. This analysis allowed discovery of an essential regulatory domain of the lipid-II flippase, MviN, that differs from the closely related pathogen *M. tuberculosis*. Given that inhibition of peptidoglycan biosynthesis is a common target for antimicrobial intervention, our identification of novel essential domains within this protein open avenues for novel therapeutic targets.

Integration of our TIS data with proteomics enabled (1) validation of expression of hypothetical proteins, including the hypothetical protein diphtheriae_00762 that appears restricted to phylogroups with the major *Corynebacterium* sp. clinical isolates, (2) identification of uncharacterized essential genes that are conserved across actinomycetia and also essential in the pathogen *M. tuberculosis*. And finally, (3) integration of our TIS data with proteomics identified current vaccine components and future vaccine target proteins. This is especially important in the context of *C. diphtheriae*, a significant global pathogen, for which there are still large gaps in our understanding of its fundamental biology. This dataset serves as an important benchmark and useful resource to the community and provides a starting point for future work.

## Materials and methods

### Bacterial strains and growth conditions

*C. diphtheriae* strain ISS 3319 is a human pathogen isolated from the throat of a 9-year old patient [103], and the strain used for this study. *C. diphtheriae* strain ISS 3319 was cultured overnight in 20 ml of Heart Infusion (HI) broth (Casein Peptone, Yeast extract, Chloride, Heart Infusion Solids; Thermo Fisher Scientific, Bremen, Germany) medium at 37˚C under shaking at 125 rpm in baffled flasks. The growth was monitored by measuring the optical density at 600 nm ($OD_{600}$) (Novapec II, Pharmacia Biotech, Uppsala, Sweden).

### Construction of the ISS3319::pK18mobDIP0053 (*pknB* mutant) strain

Standard techniques for plasmid isolation, transformation and cloning were used throughout [104]. Chromosomal disruption of the *C. diphtheriae diphtheriae_00047* (DIP0053) gene was achieved through amplification of a 549 bp internal DNA fragment of *diphtheriae_00047* using PCR from chromosomal template DNA of *C. diphtheriae* ISS 3319 using the following primers: DIP0053Mut_F 5'-aggttgcggtaaaaatgctg-3 '; DIP0053Mut_R 5'-caccctcgaaaggtggttta-3'. The PCR fragment was cloned in to pGEM-T-Easy (Promega) according to the manufacturer's instructions. The fragment was then excised from pGEM-T-Easy using *Eco*RI and sub-cloned in to dephosphorylated pK18mob cut with *Eco*RI to yield pK18mobDIP0053 [105]. This was then passaged through the non-methylating *E. coli* GM2929. One microgram of unmethylated plasmid isolated from *E. coli* GM2929 was used to transform *C. diphtheriae* using GenePulser II (Bio-Rad). Electroporated cells were allowed to recover in 1 ml of Brain Heart Infusion (BHI) broth containing 1% glucose for 2 h at 37˚C. The recovered cells were plated on to BHI medium containing kanamycin (50 μg/ml) and incubated for 18 hours at 37˚C. The plasmid pK18mob cannot replicate in *C. diphtheriae*, such that kanamycin-resistant *C. diphtheriae* are carrying the vector integrated via recombination in to the chromosomal *diphtheriae_00047*

(DIP0053) locus. The stable integration was checked by PCR using primers internal to pK18mob (pK18mob_CheckF: 5'-taggtgttagcggaccatcc-3') and a primer internal to the *diphtheriae_00047* (DIP0053) CDS (DIP0053KO_CheckR: 5'-gttttcccagtcacgacgtt-3') using genomic DNA as a template, which yields a 327bp fragment in the mutant strain, that is absent in the WT strain (S5C Fig).

### Generation of a *C. diphtheriae* transposon mutant library

The transposon library was constructed using the EZ-Tn5 Kan-2 kit (Epicentre EZI982K) and its respective instructions (S15A Fig). In short, a reaction mixture containing Transposon DNA (100 μg/ml in TE Buffer [10 mM Tris-hydrochloric acid (HCl) (pH 7.5), 1 mM EDTA]), EZ-Tn5 Transposase (1 U/μl) (Epicentre, Madison, WI, USA) and 100% glycerol was prepared for the production of stable EZ-Tn5 Transposomes. After 30 min incubation at room temperature, 1 μl EZ-Tn5 Transposome was mixed with 50 μl electrocompetent cells of *C. diphtheriae* ISS 3319 (prepared following the protocol from Ott and coauthors [62]). The mixture was incubated for 20 min on ice and used to transform *C. diphtheriae* using a GenePulser II apparatus (Bio-Rad, Munich, Germany) with 2.5 kV, 200 Ω and 25 μF set as parameters. Electroporated cells were resuspended with 1 ml of HI broth in a 1.5 ml tube, incubated in a heat block for 6 min at 46°C, transferred into 3 ml HI broth and incubated for 2 h at 37°C under shaking with 125 rpm (Edmund Bühler GmbH, Hechingen, Germany). The total volume of the culture was plated on HI agar plates containing 10 μg ml$^{-1}$ kanamycin and incubated about 48 h at 37°C. Approximately $6 \times 10^5$ colonies were pooled and stored in 15% glycerol at -80°C.

### Sample preparation for TraDIS sequencing with Illumina Miseq

The genomic DNA from 100 μl of the Tn5 mutant pool was obtained by phenol-chloroform-extraction and quantified using the dsDNA HS Qubit assay (Invitrogen). Extracted genomic DNA was fragmented by ultrasonication to an approximate size of ~250 bp and prepared for sequencing using the NEB Ultra I kit with Illumina index primers and a custom forward primer specific for the transposon (S15B Fig). Samples were quantified by qPCR using the KAPA Library Quant kit for Illumina (Kapa biosystems) and Mx3005P qPCR system (Agilent Technologies) following the kit instructions. The transposon junction was sequenced using an Illumina MiSeq v3 150 cycle cartridge.

### Processing of TraDIS sequencing data

Sequencing reads were demultiplexed first by Illumina barcodes, and then by custom 'inline index' barcodes unique to each sample replicate. Barcodes were trimmed using the fastx barcode splitter and trimmer tools [106]. The transposon was identified in a two-step pattern matching process, allowing for 3 and 1 nucleotide mismatch respectively. Reads that did not contain the transposon sequence were discarded, while reads with successful transposon-matching were trimmed of the transposon sequence and mapped to the reference genome (CP025209.1) using bwa mem [107]. The reference genome was closed and annotated by microbesNG, using Prokka (v 1.12; [108]). The plasmid deposited with this reference sequence was found to be an artefact and subsequently discarded from our analysis. Transposon insertion sequencing reads that successfully mapped to the reference genome were sorted and manipulated using samtools and bedtools [109]. The first nucleotide at the 52 end of each read was counted as the transposon insertion site. Data were viewed using the Artemis genome browser [110]. FASTQ data are available at the European Nucleotide Archive for download (accession: PRJEB56349). The processed insertion data can be viewed online at: https://tradis-vault.qfab.org/

## Calculation of essential genes

The number of unique insertion sites per gene were counted and divided by the gene length in bp to normalize for gene size, this value was termed the 'insertion index score' [48]. The frequency of insertion index scores was plotted in a histogram using the Freedman-Diaconis rule for choice of bin widths and followed a bi-modal distribution as described previously [56]. Using the R MASS library (http://www.r-project.org), an exponential distribution (red line) was fitted to the left, "essential" mode and a gamma distribution (blue line) was fitted to the right, "nonessential" mode. The probability of a gene, with a given insertion index score, belonging to each mode was calculated, and the ratio of these values was termed the 'likelihood' score.

To calculate the probability of an insertion-free region occurring by chance in a genome of a fixed size and given density, simulated libraries of stochastic transposon insertion were constructed as previously reported [56]. We used the previously reported geometric model that gives the probability of seeing k "failures" (an insertion-free site) followed by a "success" (an insertion) in a string of independent trials as $P(k) = \rho (1 - \rho)k$, where $\rho$ is the probability of an insertion. The *p*-value for a string of length L sites being insertion-free (i.e. an insertion-free region) is then $p = \sum_{k=L}^{\infty} P(k)$.

To identify genes with essential regions, we calculated the proportion of CDS that was insertion-free. We first identified all within-gene insertion-free regions with a *p*-value $\leq 0.05$; where a CDS contained more than one IFR, we summed the size of these IFRs. We then calculated the proportion of insertion-free CDS to total CDS in bp as a percentage. We used a threshold of $\geq 40\%$ to identify genes with essential regions.

## Conservation of essential genes within the *Corynebacterium* genus

To determine the conservation of 364 essential genes within the genus *Corynebacterium*, the protein sequences of these genes were searched among 140 strains representing 126 Corynebacterial species [26], using BLASTP with an E-value cut-off of $1 \times 10^{-5}$ [80]. Where more than one hit per query gene was identified, the hit with the lowest E-value was used as our focus was presence/absence of homologs. The phylogenetic relationship of *Corynebacterium* sp. were derived from a previously published dataset [26] and visualised using iTOL (itol.embl.de) [111].

## Comparison of essential genes with *M. tuberculosis* and *C. glutamicum*

Annotated genomes (.gbk) of *C. glutamicum* MB001 (Accession: NC_022040.1) and *M. tuberculosis* H37Rv (Accession: NC_000962.3) were downloaded from NCBI; gene essentiality data for these organisms were obtained from comparable transposon mutagenesis studies [47,55]. Protein fasta sequences were extracted using BBTools [112]. The essential genes of *C. diphtheriae* were compared using blastp to whole genome databases of *M. tuberculosis* and *C. glutamicum*, genes satisfying the following criteria (Evalue $\leq$1e-5, % identity $\geq$30 and % coverage $\geq$30) were classified as homologs. Gene lists were compared using BioVenn [113]. Eleven genes classified as essential in *C. glutamicum* (cgp_0693, cgp_1200, cgp_1480, cgp_1715, cgp_2093, cgp_2663, cgp_2853, cgp_3231, cgp_4102, cgp_6007) and one gene classified as essential in *M. tuberculosis* (rvnr01) were excluded from this analysis as the locus tags corresponding to these genes were not present in the respective genome annotations.

## Identification of protein domains

The coordinates of protein domains were derived from the NCBI search for conserved domains tool [91]. Transmembrane domains were predicted using phobius (https://phobius.sbc.su.se/).

## Proteome analysis

Protein preparation was carried out as previously described [90]: bacteria grown in BHI medium were harvested for whole proteome analysis and lysed with a homogenizer using glass beads (5.5 m s$^{-1}$, 30 s, 5 cycles, 4˚C). Proteins secreted into the extracellular environment were prepared as described [114]. Bacteria were removed from the culture supernatant by centrifugation (10 min, 4˚C, 4000 × *g*) and subsequently filtered using a 0.2 μm pore size filter (Minisart, Sartorius, Göttingen, Germany). Proteins were precipitated by TCA and resuspended in protein buffer (10 mM DTT, 2% sodium deoxycholate, 50 mM Tris, pH 8.0). Protein concentration was determined using Pierce 660 nm Protein Assay (Thermo Fisher Scientific, Bremen, Germany). 25 μg of the protein samples were digested using Sequencing grade trypsin carried out on 10 kDa vivacon 500 membrane filters as previously described [93,115,116]. 40 μg of the proteins were reduced and alkylated (40 mM CAA final concentration). For the in-solution digest proteins were precipitated with acetone (80% final concentration, overnight, 4˚C), resuspended in 100 mM TEAB buffer with 2 μg trypsin and digested overnight at 37˚C. Twenty-five micrograms of the resulting peptides (three biological replicates) were purified using C18 stage tips, vacuum dried, and resuspended in 0.1% trifluoroacetic acid (TFA) before being supplied for LC-MS/MS analysis [117]. Mass spectrometric analyses were carried out following a previously published protocol [90]. The separation of 10 μg of peptides were carried out by a nanoflow Ultimate 3000 HPLC (Dionex, Sunnyvale, CA, USA) using an EASY-Spray column (Thermo Fisher Scientific; C18 with 2 μm particle size, 50 cm × 75 μm) with a (flow rate of 200 nL min$^{-1}$ and increasing acetonitrile concentrations over 120 min. The total method duration including equilibration and column wash was set to 160 min. Triplicates of all samples were analyzed using an Orbitrap Fusion mass spectrometer (Thermo Fisher Scientific, Bremen, Germany) with following settings: spray voltage 2000 V, transfer tube temperature 275˚C, scan range for the MS 1 detection in the Orbitrap 300–2000 (m/z), 50 ms maximum injection time, automatic gain control (AGC) target of 4 * 10$^6$ and Orbitrap resolution of 120.000 [90]. For collision-induced dissociation with a collision energy of 35%, the ten most intense ions were selected and for ion trap detection a maximum injection time of 250 ms and an AGC target of 1 * 10$^3$ was applied. Resulting raw data files and msf files were deposited to the ProteomeXchange Consortium (http://proteomecentral.proteomexchange.org) via the PRIDE partner repository [118]. Data are available via ProteomeXchange PXD036352 (Reviewer account details: Username: reviewer_pxd036352@ebi.ac.uk; Password: BwIfWbFv). The data analyzed using the Proteome Discoverer 1.4 program package (Thermo Fisher Scientific, Bremen, Germany). A UniProt database was not available for the ISS 3319 strain but does exist for the *C. diphtheriae* reference strain NCTC 13129 (accession BX248353), therefore, we compared the annotated protein CDSs of ISS 3319 with the annotations of NCTC 13129 using blastp with the following criteria: Evalue ≤1e-5, % identity ≥30 and % coverage ≥30 (S6 Table). As described by Schäfer and co-workers [119], the theoretical masses of peptides were generated with a maximum of two missed cleavages. Due to reduction and alkylation of proteins a carbamidomethylation on cysteine was set as fixed modification. An oxidation of methionine was set as dynamic modification. To compare the measured spectra of product ions, the mass tolerance for survey scans was set to 10 ppm and 0.6 Da for fragment mass measurements. False discovery rate (FDR) was set on 1% for protein identification. Protein quantification is based on the peak area of identified proteins by using the total protein approach (TPA) [120]. Protein localization information was derived from Sangal *et al.* [58]. Proteomics data were visualized using the proteomaps program (https://bionic-vis.biologie.uni-greifswald.de/) [121–123].

### Identification of homologs in Actinobacteria species

To identify the distribution of *diphtheriae_00818* and *diphtheriae_01854* within representative Actinobacteria species BLASTP was used with an E-value cut-off of $1 \times 10^{-5}$. 51 genome sequences of representative species within the Actinobacteria phylum were obtained from NCBI and annotated using Prokka version 1.14.6 (S11 Table;[108]). The protein coding sequences of 35 genes identified as orthologous using GET_HOMOLOGUES version 20210828 [124] were aligned using MAFFT version 7.490 [125], and concatenated using AMAS [126]. A maximum-likelihood tree was generated using IQ-TREE version 1.6.12 using 10,000 ultrafast bootstraps and 10,000 SH-aLRT tests; *Bacillus subtilis* was used as an outgroup [127], and visualised using iTOL (itol.embl.de) [111]. Gene neighbourhoods were identified using webFlaGs [128].

## Supporting information

**S1 Table. Bi-modal essential gene analysis.**
(XLSX)

**S2 Table. Genes with essential regions.**
(XLSX)

**S3 Table. *Corynebacterium* sp. essential gene conservation.**
(XLSX)

**S4 Table. Conservation of *C. diphtheriae* essential gene homologs.**
(XLSX)

**S5 Table. Homologs of essential genes in *M. tuberculosis* and *C. glutamicum*.**
(XLSX)

**S6 Table. Proteomics protein function prediction.**
(XLSX)

**S7 Table. Essential protein predicted localisation.**
(XLSX)

**S8 Table. Essential protein domain prediction.**
(XLSX)

**S9 Table. Comparison of C. diphtheriae strains ISS 3319 and NCTC 13129.**
(XLSX)

**S10 Table. Proteomics data.**
(XLSX)

**S11 Table. Actinobacteria genomes.**
(XLSX)

**S1 Data. WebFlagS genes IDs to accompany S6 and S13 Figs.**
(TXT)

**S1 Fig. Method scheme: high-throughput transposon mutagenesis (TraDIS) for essential genes characterization.** Construction of a high-density transposon insertion library of 0,6 million single mutations (1), cultivation under a best condition for a bacterial library growth (2) followed by DNA extraction and sample preparation with adaptors ligation at the transposon junctions (3) for the simultaneous sequencing with a next generation sequencer (Illumina

platform) (4). The sequence data is compared with the wild type genome (5) in order to identify the essential and non-essential genes, revealing the fitness contribution of each gene under the condition analyzed (6).
(TIF)

**S2 Fig. Calculation of essential genes.** (A) Comparison of the insertion index scores of two technical replicates of the transposon mutant library. (B) Bi-modal distribution of the total insertion index scores for the transposon library. The exponential distribution fit to the left mode includes the essential genes (red), and the gamma distribution fit to the right mode captures the nonessential genes (blue).
(TIF)

**S3 Fig. Computation of the expected number and size of insertion-free regions under a null model of random insertion.** (A) The expected number of insertion-free regions (IFRs) of length l in a genome of a given size (here, 2,404,936 bp purple, and 3,079,253 bp blue, corresponding with the *C. diphtheriae* and *C. glutamicum* reference genomes used respectively) under the null model of N random insertions. The number of unique insertions was used as parameter 'N', plot A depicts the expected outcome of the null model over 100 simulations. (B) The related probability of at least one IFR of length l occuring anywhere in the simulated genome. (C and D). The same calculations repeated but within a simulated string of DNA of length g = 1,000, (equating to an IFR within a gene of 1,000 bp), repeated over $10^5$ simulations.
(TIF)

**S4 Fig. Identification of genes with an essential region.** Statistically significant insertion-free regions (IFR; > = 74 bp, pgene = 0.05) within annotated genes were calculated as a percentage of the coding sequence (CDS). Genes without a statistically significant IFR were discarded from our analysis. Where a gene had more than 1 significant IFR, the length of the IFRs were summed. The percentage of each CDS that is significantly undisrupted was calculated for each gene and coloured according to the essential classification derived from the bi-modal analysis. We applied a threshold of 40% (dashed line). Genes with >40% of the CDS undisrupted by transposon mutagenesis, but not previously identified as essential (green) were identified as domain-essential genes.
(TIF)

**S5 Fig. Conservation and essentiality of components of the PknB-MviN-FhaA phosphorylation pathway.** (A) Model of PknB phosphorylation of MviN pathway in *M. tuberculosis* adapted from Gee *et al.* (2012). (B) Transposon insertion profile of the phosphokinase *pknB* in *C. diphtheriae*. (C) PCR analysis of Wild-Type *C. diphtheriae* ISS3319 and *C. diphtheriae* ISS3319::pK18mobDIP0053 genomic DNA showing the presence of the 327 bp amplicon that indicates insertion of pK18mobDIP0053 in to the chromosome of ISS3319, which is absent from the WT strain (Markers: Lane 1: NEB 1Kb plus DNA ladder). (D) Wild-Type (WT) *Corynebacterium diphtheriae* and ISS3319::pK18mobDIP0053 ("Mutant") on BHI medium with no kanamycin (left) and kanamycin selection (50 μg/ml) (right) on BHI medium indicating stable insertion of pK18mobDIP0053 in to the chromosome. (E)Transposon insertion data of *fhaA* capped at a frequency of 1. Protein domains are drawn beneath the gene track. (F) (i) Sequence comparison between Cdip-FhaA and Mtb-FhaA with both the domain of unknown function, DUF3662, and Fork-head associated (FHA) domain highlighted in coloured boxes. The conserved Ser[473] and Arg[474] of Mtb-FhaA that interact with P-Thr[97] of MviN are highlighted with an asterisk (*). (ii) A prediction of the Cdip-FhaA structure with the DUF3662 and FHA domains highlighted. (G) (i) The solved structure of the FHA domain of Mtb-FhaA (green) interacting with the cytoplasmic domain of Mtb-MviN (grey; PDB: 3OUN). (ii) A model of Cdip-FhaA FHA (amber) alongside Mtb-MviN, and (iii) an overlay of both Cdip- and Mtb-

FHA domains for structural comparison.
(TIF)

**S6 Fig. Gene neighbourhood of *mviN* orthologs within Actinobacteria species.** Gene neighbourhood of *diphtheriae_02308 (mviN)* orthologs (black) in representative Actinobacteria species, figure generated using FlaGs. Gene (6) is annotated as a "protein kinase family protein" and is frequently observed downstream from *mviN* orthologs that do not have an extended sequence. The remaining predicted gene functions associated with each number are provided in Supplementary Information.
(TIF)

**S7 Fig. Conservation of essential genes across the *Corynebacterium* genus.** Heatmap showing the presence/absence of protein orthologs of the 358 essential genes (*x*-axis) identified in *C. diphtheriae* in the genomes of 140 representative *Corynebacterium* species (*y*-axis) adapted from Dover *et al.* (2021). Blocks are coloured blue according to percentage identity, above a threshold of 30%, with darker shading corresponding with a higher percentage identity shared with the *C. diphtheriae* query gene.
(PDF)

**S8 Fig. Conservation of *isp* and *dap* genes, and the *desA3* operon, across the *Corynebacterium* genus.** Homologs were identified by a blastp search and coloured in blue according to the percentage identity to the respective query (*C. diphtheriae* ISS 3319) gene. Note "*dapL*" is likely a mis-annotation of "*dapC*" based on protein alignment analyses.
(PDF)

**S9 Fig. Essential gene pathways that are not fully conserved across the *Corynebacterium* genus.** The diaminopimelate biosynthesis pathway and the methyl erythritol phosphate pathway of isoprenoid synthesis. The genes encoding the enzymes required for each step are coloured in blue.
(TIF)

**S10 Fig. FtsB homology and essential domains.** (A) Transposon insertion frequency within the *mrp* operon, with essential genes and regions highlighted in teal. (B) (i) The gene neighbourhood of *ftsB* (*ftsL_1; diphtheriae_00817*) homologs (black), figure generated by webFlaGs. The numbered genes encode: 1. phosphopyruvate hydratase; 2. Ppx/GppA family phosphatase; 3. nucleotide pyrophosphohydrolase; 4. lytic murein transglycosylase; 5. DUF501 domain-containing protein. (ii) The gene neighbourhood of *ftsL* (*ftsL_2; diphtheriae_01542*) homologs (black), figure generated by webFlaGs. The numbered genes encode: 1. UDP-N-acetylmuramoyl-L-alanine—D-glutamate ligase; 2. division/cell wall cluster transcriptional repressor MraZ; 3. rRNA small subunit methyltransferase H RsmH; 4. penicillin-binding membrane protein; 5 phospho-N-acetylmuramoyl-pentapeptide-transferase.
(TIF)

**S11 Fig. Predicted localization of essential proteins.** The predicted localization of essential proteins adapted from Sangal *et al.* (2015) shown for the total proteome (top), the predicted transmembrane proteins (left, 51) and predicted secreted proteins (right, 25). Abbreviations: Cyt, cytoplasmic; TM, transmembrane; Sec, secreted; Lipo, lipoprotein; NC, non-classical secreted protein; Spi, SpI type signal peptide; Tat, Tat signal peptide; Amb, ambiguous.
(TIF)

**S12 Fig. Conserved essential genes restricted to actinomycetes.** (A) and (B) Transposon insertion data for *diphtheriae_00818* and *diphtheriae_01854* genes with the respective domains

of unknown function (DUF) DUF501 (PF04417) and DUF3073 (PF11273) displayed beneath. Transposon insertion sites are represented by vertical black bars, capped at a frequency of 1. (C) Distribution of homologs of diphtheriae_00818 and diphtheriae_01854 within representative genomes of the Actinobacteria phylum. *Bacillus subtilis* was used as an outgroup for construction of the tree; only bootstrap values under 100 are shown on the tree. Species are shaded by Class. The presence of a homolog, identified by BLASTP, is indicated by a coloured circle. (TIF)

**S13 Fig. Homologs of *diphtheriae_00818* and *diphtheriae_01854*.** (A) Protein sequence alignment of the essential homologs of (i) *diphtheriae_00818* and (ii) *diphtheriae_01854* conserved in *M. tuberculosis* and *C. glutamicum*. Alignments generated using EMBL-EBI MUSCLE, conserved residues are shaded. (B) Gene neighbourhood of *diphtheriae_00818* homologs (shown in black) in representative Actinobacteria species, figure generated using FlaGs. The remaining predicted gene functions associated with each number are provided in Supplementary Information. (TIF)

**S14 Fig. Essential secreted proteins detected in the diphtheria vaccine.** (A) Transposon insertion frequency within the genes of essential proteins detected in the secreted fraction. Essential genes and regions are highlighted in teal. Transposon insertion sites are represented by black bars and capped at a frequency of 1. Protein domains were predicted using the NCBI's conserved domain (CD-) search [91], and are displayed beneath the gene tract. (B) Sequence alignment of the amino acids of diphtheriae_02141 and PS1 from *C. glutamicum* (WP_011015455.1), figure generated using ESPript [129]. The conserved catalytic triad are highlighted with (*). (TIF)

**S15 Fig. Details of the transposon and sequencing preparation.** (A) Schematic of the commercially available mini EZ-Tn*5* transposon. (B) Schematic of the PCR step that introduces the necessary barcodes and sequences for Illumina sequencing. An inline index barcode is introduced to differentiate between samples and to stagger the start of the transposon sequence during amplicon sequencing. Following successful identification of the inline index barcode during sequence data analysis, the transposon sequence is identified in two steps: Tntag1 (blue) and Tntag2 (amber), which correspond with the primer binding site and the remaining transposon sequence immediately downstream, respectively. Discrepancies between Tntag1 and Tntag2 reveal mis-priming errors during sample preparation for sequencing; these tags correspond with"transposon check 1" and "transposon check 2" described in Table 1. (TIF)

## Acknowledgments

We thank Emily Richardson and microbesNG for sequencing the original reference strain and assistance with annotating and interpreting the reference genome. We thank Tim Bruxner and Angi Christ at the IMB Sequencing Facility for an excellent service generating the PacBio sequencing data.

## Author Contributions

**Conceptualization:** Emily C. A. Goodall, Camila Azevedo Antunes, Jens Möller, Andreas Burkovski, Ian R. Henderson.

**Data curation:** Emily C. A. Goodall, Camila Azevedo Antunes, Jens Möller, Vartul Sangal, Jessica Gray.

**Formal analysis:** Emily C. A. Goodall, Camila Azevedo Antunes, Jens Möller, Vartul Sangal, Von Vergel L. Torres, Jessica Gray.

**Funding acquisition:** Paul A. Hoskisson, Andreas Burkovski, Ian R. Henderson.

**Investigation:** Emily C. A. Goodall, Camila Azevedo Antunes, Jens Möller, Vartul Sangal, Von Vergel L. Torres.

**Methodology:** Emily C. A. Goodall, Camila Azevedo Antunes, Jens Möller.

**Project administration:** Camila Azevedo Antunes.

**Resources:** Vartul Sangal, Adam F. Cunningham, Paul A. Hoskisson, Andreas Burkovski, Ian R. Henderson.

**Software:** Emily C. A. Goodall, Jessica Gray.

**Supervision:** Paul A. Hoskisson, Andreas Burkovski, Ian R. Henderson.

**Validation:** Vartul Sangal, Paul A. Hoskisson.

**Visualization:** Emily C. A. Goodall, Camila Azevedo Antunes, Jens Möller, Von Vergel L. Torres, Paul A. Hoskisson.

**Writing – original draft:** Emily C. A. Goodall, Camila Azevedo Antunes, Jens Möller.

**Writing – review & editing:** Vartul Sangal, Von Vergel L. Torres, Jessica Gray, Adam F. Cunningham, Paul A. Hoskisson, Andreas Burkovski, Ian R. Henderson.

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
