## [Decision Letter · Decision Letter 0]

24 Jan 2023

Dear Dr Goodall,

Thank you very much for submitting your Research Article entitled 'A multiomic approach to defining the essential genome of the globally important pathogen *Corynebacterium diphtheriae*' to PLOS Genetics.

The manuscript was fully evaluated at the editorial level and by independent peer reviewers. The reviewers appreciated the attention to an important problem, but raised some substantial concerns about the current manuscript. Based on the reviews, we will not be able to accept this version of the manuscript, but we would be willing to review a much-revised version. We cannot, of course, promise publication at that time.

Should you decide to revise the manuscript for further consideration here, your revisions should address the specific points made by each reviewer. Among the many good suggestions from the reviewers to improve your paper, reviewer #3 requested genetic validation of at least some of the inferences drawn from the TnSeq data - if you can do this it would greatly strengthen the significance of your findings. We will also require a detailed list of your responses to the review comments and a description of the changes you have made in the manuscript.

If you decide to revise the manuscript for further consideration at PLOS Genetics, please aim to resubmit within the next 60 days, unless it will take extra time to address the concerns of the reviewers, in which case we would appreciate an expected resubmission date by email to plosgenetics@plos.org.

We are sorry that we cannot be more positive about your manuscript at this stage. Please do not hesitate to contact us if you have any concerns or questions.

Yours sincerely,

Diarmaid Hughes

Academic Editor

PLOS Genetics

Lotte Søgaard-Andersen

Section Editor

PLOS Genetics

Reviewer #1: In this article the authors use transposon insertion sequencing, proteomics and data analysis to determine which are the essential genes in Corynebacterium diphtheriae. They also explore the distribution of these genes within the Corynebacterium genus, the presence of the essential proteins in the Diphtheriae vaccine, and they compare the essential genome of C. diphtheriae with those of Mycobacterium tuberculosis and Corynebacterium glutamicum.

The approaches used in the article are appropriate, the results presented are novel, the article is well written, and it is of great interest not only to the community that studies diphtheria, but also to the community that studies different members of Corynebacteriales.

However, I believe that some points should be addressed before publication:

Line 163. How did you choose the threshold of 12? Is it arbitrary? It should be stated.

Line 169. What does it mean to fall in the category “unclear” from a biological point of view? Is it really “unclear” the word that describes this group? I guess that, for example, some of these proteins are not essential but the mutants are more sensitive to the environment. I think this should be discussed.

Line 188. I think that the approach you used to identify essential “regions” is correct. But why not also identify essential “domains”? You could have used the NCBI conserved domains tool on the total proteins to delimit the domains, and then the same approach you used for the identification of the essential proteins, in this case on each domain. This would be useful especially in proteins with more than 2 domains, or when the domains have very different lengths. This is just a suggestion, but I really think it would enrich the article. I believe that this would be an incredible contribution to the Corynebacteria community, in particular, for determining the importance of domains of unknown function.

Line 197. I’m not sure if I understood the approach. Should it say “calculating the proportion of insertion-free regions” instead of “calculating the proportion of insertion-free CDS”?

Line 261. What are the annotation artifacts that you mention?

Line 290. It would be useful to indicate which are the groups of genes with differential conservation on Supplementary Figure 7.

Line 291. What do you think it means that some genes of the isoprenoid biosynthetic pathway are present but others are absent in some species? Are these genes shared with other pathways? Is it possible that you did not identify the genes because of the approach you used or because the genomes are not complete?

Line 338. In order to check the absences of the essential genes in Mtb you should try more sophisticated approaches than blastp, like Dali searches against the Alphafold database (http://ekhidna2.biocenter.helsinki.fi/dali/). I tried it for a couple of the atp proteins you did not identify using blastp and I obtained good results.

Line 336. I think that you should also review the localization of the total of the proteins and compare these results with those of the essential ones. Membrane proteins in Corynebacteriales are not well characterized, so it would be very interesting to know what proportion of them are essential proteins.

Line 391. Which are the essential proteins not detected by the proteome? You mention that many are associated by cell envelope but I could not find this information. For the proteins that are not associated with the cell envelope, what are the reasons why you may not have detected them in the proteomics analysis? You only mention silencing by DNA-binding proteins, but you should mention the limitations of the approach considering that you did not detect a large fraction of the essential proteins.

Line 407. In Table 3, why did you present the abundance of the essential proteins as a fraction of the total? I am not sure it is the best format. I consider it is very important to show that more than half of the essential secreted proteins are uncharacterized, and this information is diluted when you present it in that format.

Line 447. In order to provide some context, I think you should mention what is the total amount of proteins present in the diphtheria vaccines you are referring to.

Line 537. You mention that the plasmid was an artifact. Can you explain how you determined that?

Line 579. .gbk files are not provided anymore. I guess you meant .gff files.

Line 633. I would appreciate it if you could comment on the differences you found in the comparison between strains ISS3319 and NCTC 13129. I guess most of them are associated with the synthesis of the toxin, but I’m curious about it.

Line 652. Did you check if the genes identified as orthologous can actually be concatenated? Did you do any topology test before concatenation?

Line 658. In the data availability statement you indicate that the annotated genome can be accessed at the ENA site, but I could not find this information. It is extremely important that you provide the annotations of the genome so the community can make use of the data you generated in this article. It is also of huge importance that every time that you mention a gene like “diphtheriae_XXXXX” you also include the corresponding DIP gene tag, otherwise it is very hard to know which is the gene you are referring to.

Supplementary Figures 6 and 12. Which are the genes indicated with numbers? These numbers should be specified in the legend or not provided at all.

Reviewer's Responses to Questions

**Comments to the Authors:**

Reviewer #1: In this article the authors use transposon insertion sequencing, proteomics and data analysis to determine which are the essential genes in Corynebacterium diphtheriae. They also explore the distribution of these genes within the Corynebacterium genus, the presence of the essential proteins in the Diphtheriae vaccine, and they compare the essential genome of C. diphtheriae with those of Mycobacterium tuberculosis and Corynebacterium glutamicum.

The approaches used in the article are appropriate, the results presented are novel, the article is well written, and it is of great interest not only to the community that studies diphtheria, but also to the community that studies different members of Corynebacteriales.

However, I believe that some points should be addressed before publication:

Line 163. How did you choose the threshold of 12? Is it arbitrary? It should be stated.

Line 169. What does it mean to fall in the category “unclear” from a biological point of view? Is it really “unclear” the word that describes this group? I guess that, for example, some of these proteins are not essential but the mutants are more sensitive to the environment. I think this should be discussed.

Line 188. I think that the approach you used to identify essential “regions” is correct. But why not also identify essential “domains”? You could have used the NCBI conserved domains tool on the total proteins to delimit the domains, and then the same approach you used for the identification of the essential proteins, in this case on each domain. This would be useful especially in proteins with more than 2 domains, or when the domains have very different lengths. This is just a suggestion, but I really think it would enrich the article. I believe that this would be an incredible contribution to the Corynebacteria community, in particular, for determining the importance of domains of unknown function.

Line 197. I’m not sure if I understood the approach. Should it say “calculating the proportion of insertion-free regions” instead of “calculating the proportion of insertion-free CDS”?

Line 261. What are the annotation artifacts that you mention?

Line 290. It would be useful to indicate which are the groups of genes with differential conservation on Supplementary Figure 7.

Line 291. What do you think it means that some genes of the isoprenoid biosynthetic pathway are present but others are absent in some species? Are these genes shared with other pathways? Is it possible that you did not identify the genes because of the approach you used or because the genomes are not complete?

Line 338. In order to check the absences of the essential genes in Mtb you should try more sophisticated approaches than blastp, like Dali searches against the Alphafold database (http://ekhidna2.biocenter.helsinki.fi/dali/). I tried it for a couple of the atp proteins you did not identify using blastp and I obtained good results.

Line 336. I think that you should also review the localization of the total of the proteins and compare these results with those of the essential ones. Membrane proteins in Corynebacteriales are not well characterized, so it would be very interesting to know what proportion of them are essential proteins.

Line 391. Which are the essential proteins not detected by the proteome? You mention that many are associated by cell envelope but I could not find this information. For the proteins that are not associated with the cell envelope, what are the reasons why you may not have detected them in the proteomics analysis? You only mention silencing by DNA-binding proteins, but you should mention the limitations of the approach considering that you did not detect a large fraction of the essential proteins.

Line 407. In Table 3, why did you present the abundance of the essential proteins as a fraction of the total? I am not sure it is the best format. I consider it is very important to show that more than half of the essential secreted proteins are uncharacterized, and this information is diluted when you present it in that format.

Line 447. In order to provide some context, I think you should mention what is the total amount of proteins present in the diphtheria vaccines you are referring to.

Line 537. You mention that the plasmid was an artifact. Can you explain how you determined that?

Line 579. .gbk files are not provided anymore. I guess you meant .gff files.

Line 633. I would appreciate it if you could comment on the differences you found in the comparison between strains ISS3319 and NCTC 13129. I guess most of them are associated with the synthesis of the toxin, but I’m curious about it.

Line 652. Did you check if the genes identified as orthologous can actually be concatenated? Did you do any topology test before concatenation?

Line 658. In the data availability statement you indicate that the annotated genome can be accessed at the ENA site, but I could not find this information. It is extremely important that you provide the annotations of the genome so the community can make use of the data you generated in this article. It is also of huge importance that every time that you mention a gene like “diphtheriae_XXXXX” you also include the corresponding DIP gene tag, otherwise it is very hard to know which is the gene you are referring to.

Supplementary Figures 6 and 12. Which are the genes indicated with numbers? These numbers should be specified in the legend or not provided at all.

Reviewer #2: Goodall et al. constructed and analyzed a high-density transposon directed insertion library in C. diphteria to identify the essential genome. The authors find 341 essential genes that largely correspond to essential genes of other species within the Actinobacteriota.

This work contains an impressively dense library and a very thorough analysis. I only have a few questions that are mainly related to the EZ-Tn5 Kan-2 kit that is used in the study. I would appreciate a couple of sentences and maybe a small supplementary figure what the effect of an insertion within a coding sequence is (i.e., How large is the insertion? Does it contain outward facing promotors?).

1. How does the insertion affect operon structures? Is it possible that a non-essential gene is classified as essential because the insertion disrupts transcription of another gene within the operon?

2. Can you detect essential RNAs?

3. Are there duplicate genes in the genome that are potentially essential but would be missed by this approach since either can the deleted individually?

4. Lines 166-169: Have the 115 genes that are classified as unclear been analyzed further? I agree that the insertion density within the resA_2 gene (Fig 1) seems lower than for the others but based on the insertion pattern resA_2 doesn’t look like an essential gene to me. Also, the gene name indicates that there are more copies of this gene on the genome. Could that affect the results?

5. Is there a sequence bias for insertion? Could some of the ‘unclear’ genes be explained due to this bias?

6. Lines 196-201: Does this analysis for essential domains only work for the N-terminus of a gene or also for the C-terminus?

7. Lines 202-253: Would the insertion of a transposon in the middle of the gene split the protein into two independent proteins? A MurJ-like protein and a Pseudokinase protein.

8. Fig 1C: The figure looks like transposons are inserted right up to the start codon of nadE. Does the Tn5 transposon contain a transcriptional start and ribosome binding site?

9. How are occasional insertions within essential genes explained (i.e., mrpA in Fig S10)?

Reviewer #3: This paper from Goodall and co-workers describes the use of transposon insertion sequencing (TnSeq/TraDIS) to characterize the essential genome of the pathogen Corynebacterium diptheriae (Cdip). This bacterium belongs to the Corynebacteriales order that includes other pathogens like Mycobacterium tuberculosis (Mtb) and model organisms like Mycobacterium smegmatis and Corynebacterium glutamicum (Cglu). These bacteria have a unique cell envelope consisting of multiple layers including the peptidoglycan (PG), arabinogalactan (AG), and mycomembrane (MM). Unlike most model bacteria, they also grow via tip extension. Thus, understanding the biology of these bacteria is important for addressing fundamental questions about cell growth as well as for practical reasons of therapeutic development.

The main advance of this paper is the generation of the first high-density transposon library in a pathogenic Corynebacterium and using it to determine its essential genome and compare it to other results from Mtb and Cglu. The paper is very well written and the bioinformatics is well done. The dataset generated will be highly valuable to the field.

Major critique:

Where the paper fall a little short in my opinion is the lack of genetic validation of any of the inferences drawn from the TnSeq data. The results would be much stronger if some of the observations were followed up with genetic experiments. For example, the authors claim that pknB is non-essential in Cdip unlike Mtb and that the mrp operon is essential and a unique transport pathway distinct from Mtb. My understanding is that tools are available to engineer Cdip, so adding some genetic validation to strengthen the results should be feasible.

Minor points:

1) The tables require more detail in the legends. It is not clear what “inline barcode match”, “transposon check 1 or 2” refer to in Table 1. Insertions density should be defined in Table 2 legend.

2) For the predictions of secreted proteins, it would be nice to add Tat transport and sortase signals.

**Have all data underlying the figures and results presented in the manuscript been provided?**

Reviewer #1: **No: **The genome annotation file is not provided.

Reviewer #2: Yes

Reviewer #3: Yes

PLOS authors have the option to publish the peer review history of their article (what does this mean?). If published, this will include your full peer review and any attached files.

Reviewer #1: **Yes: **Daniela Megrian

Reviewer #2: No

Reviewer #3: No

---

## [Decision Letter · Decision Letter 1]

8 Apr 2023

Dear Dr Goodall,

We are pleased to inform you that your manuscript entitled "A multiomic approach to defining the essential genome of the globally important pathogen *Corynebacterium diphtheriae*" has been editorially accepted for publication in PLOS Genetics. Congratulations!

Yours sincerely,

Diarmaid Hughes

Academic Editor

PLOS Genetics

Lotte Søgaard-Andersen

Section Editor

PLOS Genetics

Comments from the reviewers (if applicable):

Reviewer's Responses to Questions

**Comments to the Authors:**

Reviewer #1: I believe that the authors addressed most of my concerns and enriched the article accordingly. I have just a few minor comments (line numbers refer to the numbers in the author's answers):

Line 169. The references are missing in the text that was added.

Line 338. While conserved structural fold is not necessarily evidence for a functional analogue, relying on blastp hits is definitely much worse. There were not many absences of essential genes to verify, I believe you should have used more reliable approaches.

Line 579. You did used Genbank files, but those were not ".gbk" files as stated, because that extension has been deprecated by the NCBI several years ago (https://www.ncbi.nlm.nih.gov/genome/doc/ftpfaq/).

Line 652. The answer does not reflect what was done in the article. In the materials and methods section "Identification of homologs in Actinobacteria species", you explain thay you obtained 35 orthologous genes that you aligned and concatenated to reconstruct a reference phylogeny of Actinobacteria. You should have verified that those genes were suitable for concatenation (i.e. congruence analysis), or at least justify why you didn't verify. In this case, I would say that the topology you obtained resembles the Actinobacteria phylogenies previously published, and that would be enough considering you are just using it to present a phyletic pattern.

Reviewer #2: The authors have addressed all of my comments.

Reviewer #3: The authors have nicely addressed my concerns and those of the other reviewers.

**Have all data underlying the figures and results presented in the manuscript been provided?**

Reviewer #1: Yes

Reviewer #2: Yes

Reviewer #3: Yes

PLOS authors have the option to publish the peer review history of their article (what does this mean?). If published, this will include your full peer review and any attached files.

Reviewer #1: **Yes: **Daniela Megrian

Reviewer #2: No

Reviewer #3: No

**Data Deposition**

http://datadryad.org/submit?journalID=pgenetics&manu=PGENETICS-D-22-01369R1

**Press Queries**

---

## [Editor Report · Acceptance letter]

21 Apr 2023

PGENETICS-D-22-01369R1 

A multiomic approach to defining the essential genome of the globally important pathogen *Corynebacterium diphtheriae*

Dear Dr Goodall, 

We are pleased to inform you that your manuscript entitled "A multiomic approach to defining the essential genome of the globally important pathogen *Corynebacterium diphtheriae*" has been formally accepted for publication in PLOS Genetics! Your manuscript is now with our production department and you will be notified of the publication date in due course.

With kind regards,

Zsofi Zombor

PLOS Genetics

On behalf of:
